# Examining the Role of Service Quality, Perceived Values, and Trust in Macau Food Festival

**DOI:** 10.3390/ijerph18179214

**Published:** 2021-09-01

**Authors:** Chen-Kuo Pai, Tingting Lee, Sangguk Kang

**Affiliations:** 1Faculty of Hospitality and Tourism Management, Macau University of Science and Technology, Aveida Wai Long, Taipa, Macau, China; ckpai@must.edu.mo (C.-K.P.); 2009853lbt30002@student.must.edu.mo (T.L.); 2Department of Tourism Management, College of Social Sciences, Gangneung-Wonju National University, Jukheon-gil 7, Gangneung 25457, Korea

**Keywords:** Macau food festival, festival quality, festival value, trust in festival, visitors’ satisfaction, revisit intention

## Abstract

An emerging paradigm for festival and event management reveals that hosting theme-based festivals can foster economic growth at the local and national levels. As a case of this research, the annual Macau Food Festival was selected to explore relationships among visitors’ perceptions of festival quality, festival value, trust in the festival, visitors’ satisfaction, and behavior intention. Out of 453 questionnaires distributed, 433 questionnaires were valid for data analysis using partial least squares-structural equation modeling. The results showed that festival quality consisting of the sub-dimensions of hospitality, venue, product, convenience, and program have a positive impact on festival value, trust in festival, and festival satisfaction, which in turn affect festival loyalty. Based on these empirical findings, the current study offers practical and theoretical implications for destination management organizations and festival hosts to sustain annual theme-based festivals held at a certain destination such as Macau Food Festival.

## 1. Introduction

An emerging paradigm for festival and events management reveals that hosting theme-based festivals enables not only fostering economic growth at the local and national levels [1,2] but also shaping the positive image of hosting areas [1], which consequently aims to facilitate the level of festival loyalty. As an index for predicting future demand for an annual theme-based festival, exploration of how festival visitors form festival loyalty in a cognitive process has been highlighted by many scholars due to its critical role in detecting if festival visitors are willing to be repeat visitors [3]. The necessity of understating festival loyalty calls for a deeper investigation of what factors will directly and indirectly affect festival loyalty in a cognitive process model. Given this recognition, many studies have proposed the prominent attributes of service quality and identified their impacts on loyalty [4].

The research literature of festival management seldom talks about a new set of festival quality and its relationship with festival loyalty in a cognitive process model. To better understand the function of theme-based festivals in the formation of visitors’ loyalty, destination management organizations (DMOs) and festival hosts are urged to recognize the prominent attributes of festival quality enabling festival attendees to perceive higher levels of festival value [5], trust in the festival [6,7,8], satisfaction with the festival [8,9,10], and festival loyalty [10,11,12] in a festival hosting destination. This reveals that exploring a set of attributes of festival quality (i.e., hospitality, venue, product convenience, and program) is important for festival hosting destinations (e.g., DMOs) to sustain theme-based festivals and promote festival hosting destinations in a competitive market [8,13,14]. Therefore, this study attempts to identify salient quality attributes of theme-based festivals (e.g., the annual Macao Music Festival), which helps overcome the drawback of the SERVQUAL research in festival settings.

In this regard, an in-depth literature review of theme-based festival attributes yields the following research question: how does festival quality (including salient attributes) affect festival loyalty in a cognitive process model? Therefore, the current study aims to explore the relationships among festival quality (five sub-dimensions of hospitality, venue, product convenience, and program), festival value, trust in the festival, satisfaction with the festival, and festival loyalty. The results offer practical and theoretical implications for destination management organizations (DMOs) and festival hosts to develop a competitive theme-based festival design and sustain annual theme-based festivals held at a certain destination (e.g., the Macau Food Festival).

## 2. Literature Review

### 2.1. The Role of Theme-Based Festivals

In the current trend of international relations and economic development, Macao, as an international city where Chinese and Western cultures intersect, promotes and highlights its tourism destination image by hosting various festivals and events, and it attracts tourists from all over the world. According to the Macao Tourism Bureau, there were 23 events in Macao in 2019, including the annual Macao food festival. In 2010, Macao was awarded the honorary title of “the world culinary capital” by UNESCO (United Nations Educational, scientific and Cultural Organization). Since 2000, the annual Macao food festival has been successfully held 19 times. The 19th Macao Food Festival had the theme “Celebrate the 20th anniversary of Macau reunification with world gourmet”, and invited more than 150 famous catering companies to demonstrate the unique multi-cuisine culture of Macau and present a feast of “the world culinary capital” for tourists. Festival and special events can be divided into subsectors such as traditional sacrifice, exhibition, festival, conference, art and culture, physical and sports, music, regional dispersion, and landscape performance (e.g., [15,16,17]). However, recent studies have pointed out the need to target tourists who are interested in unique, authentic, and valuable experiences in favor of the themes of music, food, wedding, heritage, etc. in the context of community-organized annual festivals [18,19]. Given the fact that theme-based festivals are effective platforms for providing the opportunity for tourists to experience theme-based products and services in festival hosting destinations, some studies have focused on the role of theme-based festivals in local, regional, and national destinations. According to Lee et al. [14], the advantages of theme-based festivals (music, heritage, etc.) encompass (a) unique experiences for staycation travelers (authentic food, music, crafts, etc.), and (b) theme-based entertainment activities (e.g., quizzes for regional stories), and (c) revitalizing the economy of festival-hosting destinations. Therefore, hosting theme-based festivals enables DMOs and festival hosts to not only enhance economic growth but also draw frequent visitors to the festival hosting destinations.

### 2.2. Festival Quality

From a psychological perspective, Gronroos [20] coined the concept of perceived service quality as an index for detecting the degree of personal perceptions toward service experiences. Based on this theoretical foundation, the SERVQUAL scale [21] and the SERVPERF (five dimensions: tangibles, reliability, responsiveness, assurance, and empathy) scale [22] were developed and utilized to evaluate the level of expected service experiences in hospitality and tourism research. However, it would not be possible for researchers to measure the magnitude of service quality for festivals using the two scales because individuals’ expected experiences at a special interest festival would be more detailed than at other service experiences from hotels, restaurants, and destinations (e.g., [23]). In this regard, some studies have verified that the scale of service quality for events including hospitality, venue, product, convenience, and program should be employed to predict visitors’ cognitions such as perceived value, trust, and satisfaction in the formation of festival loyalty [7,8,24]. In this regard, theme-based festival quality can be measured by the attributes of hospitality, venue, product, convenience, and program in festival settings. The concepts of each festival quality utilized in this study are summarized as follows.

#### 2.2.1. Hospitality

Hospitality, the first sub-dimension of festival quality, includes courtesy, behavior, and business ability of employees at festivals and special events [17]. The hospitality attribute directly affects tourists’ perception of the service quality of theme-based festivals, which in turn affects their satisfaction [8]. Lee et al. [25] conducted a survey for gauging the level of tourists’ perceptions toward the service quality of international festivals. The results implied that hospitality should be a significant component of festival quality, which indirectly affects festival satisfaction and loyalty. More importantly, Song et al. [7] found that hospitality is one of the important measurement dimensions of service quality for events and concluded that hospitality is an important predictor of tourist satisfaction as a critical attribute of festival quality.

#### 2.2.2. Venue

As the second sub-dimension of festival quality, venue refers to the place where festivals and special events are planned to be carried out [8], including scale, environment, and on-site atmosphere [26]. A number of studies regard venue as an important sub-dimension of festival quality in determining the degree of satisfaction, trust, and loyalty [7]. Importantly, Lee et al. [25] highlighted that the location of festival events (such as on-site public facilities, site size, site cleanliness, and site atmosphere) has a positive relationship with satisfaction. In this vein, Shonk and Chelladurai [26] regarded the venue as one of the significant dimensions enhancing the magnitude of festival quality. They also proposed that venue affects tourists’ perceived value, which in turn facilitates the levels of satisfaction and loyalty.

#### 2.2.3. Product

A number of studies have proposed product as one of the significant dimensions measuring festival quality. As the third sub-dimension of festival quality, product refers to foods, beverages, and souvenirs that can be purchased at a theme-based festival (e.g., the Macau Food Festival). Yoon, Lee, and Lee [27] explored tourists’ perceptions about festival quality dimensions such as program, souvenirs, food, and facilities in the context of the Punggi Ginseng festival. They found that the attributes of tangible products (including souvenirs) might allow festival visitors to perceive a positive sense of value, which in turn enhances the degrees of satisfaction and loyalty. Importantly, Lee, Lee, and Choi [28] demonstrated that product should be utilized as a critical factor of event quality when it comes to its positive impact on tourist satisfaction. Furthermore, Anil [29] verified that the festival’s environmental factors such as food, festival area, and convenience influence the visitors’ satisfaction and, consequently, their loyalty.

#### 2.2.4. Convenience

Convenience, the fourth sub-dimension of festival quality, refers to the time and effort that customers are willing to save when purchasing or using a certain product or service [30]. Given this definition, convenience can be utilized as an index for measuring the extent to which individuals can use available facilities or goods offered at a convenient time and place [31]. From a theoretical perspective, many scholars have pointed out the significance of convenience as a key precursor to tourist loyalty in festival settings. For example, Lin and Lee [32] confirmed that the convenience of festival activities plays a critical role in improving the quality of theme-based festivals (e.g., cultural festivals). Lee et al. [25] also conducted an on-site survey of 500 tourists participating in large-scale international festivals and found that convenient transportation and food quality may affect tourist satisfaction and ultimately their loyalty. More importantly, Lee et al. [28] conducted an empirical study on Korea’s Boryeong Mud Festival and verified that convenience is an important factor affecting tourist satisfaction and loyalty in festival settings. Altogether, it is necessary to understand whether the festival site has convenient conditions or facilities in the context of theme-based festivals.

#### 2.2.5. Program

Program is regarded as the fifth sub-dimension of festival quality. Smith and Forest [33] posit that the program should strike a balance between expression and entertainment functions, tourist needs and local residents, and cultural communication and cultural protection. A high-quality event program allows tourists to perceive a positive sense of festival at a special event, which may enhance the magnitude of their satisfaction and loyalty [25,27]. More importantly, Cole and Chancellor [17] proposed the salient dimensions (i.e., program, entertainment facilities, entertainment quality) of festival quality. They implied the need for a program as a facilitator of visitors’ overall experience in festival settings. Consistent with this, Lee, Lee, and Yoon [34] also demonstrated that program is a critical dimension of festival quality, which plays an important role in fostering the degree of festival value and satisfaction.

### 2.3. Festival Value

Value refers to the overall evaluation after comparing the cost paid by customers in obtaining products or services with their perceived value [21]. Generally speaking, perceived value is generated by the exchange of revenue and cost perceived by customers [35]. Cronin, Brady, and Hult [36] investigated the impact of service quality for event, customers’ perceived value, and satisfaction on loyalty and confirmed that customers’ perceived service quality has a significant impact on perceived value. Chen and Chung [37] investigated online shopping customers in Korea by a questionnaire and also confirmed the positive impact of service quality for events on customers’ perceived value. It can be seen that there is a direct relationship between perceived value and service quality for an event. When the perceived service quality of tourists is totally satisfied, their perceived value will also increase synchronously. Therefore, this study proposes the following hypothesis:

**Hypothesis** **1** **(H1).**
*Festival quality has a positive impact on festival value.*


### 2.4. Trust in Festival

Andaleeb and Anwar [38] think that trust refers to a trustor being willing to believe and rely on the trustee in a specific situation; Fam, Foscht, and Collin [39] pointed out that trust is a dynamic process instead of static, and it takes a period to establish, so as to help customers build a satisfaction beyond the economic effect. Gronin et al. [38] found that the service quality for event has a positive impact on the customers’ perceived value and trust, and the service quality for the event also directly affects the customers’ behavior intention. Chaudhuri and Holbrook [40] found that perceived value has a positive impact on trust, and trust will also positively affect loyalty. However, Hsieh and Liu [41] found that perceived value has no direct impact on customers’ trust. To sum up, it can be seen that there is a correlation between the service quality for event and trust. When tourists are fully satisfied with the perceived service quality for event, their trust will also rise synchronously. However, there are different conclusions in the research on perceived value and trust, so it is necessary to investigate their relationship. Therefore, this study proposes the following hypotheses:

**Hypothesis** **2** **(H2).**
*Festival quality has a positive impact on trust in the festival.*


**Hypothesis** **3** **(H3).**
*Festival value has a positive impact on trust in the festival.*


### 2.5. Satisfaction with Festival

Cardozo [42] first pointed out that customer satisfaction has an important impact on customers’ purchase behavior. According to Bolton and Drew [43], customers’ satisfaction refers to all the emotions generated by customers in the whole purchasing process, which will affect their behavioral intentions in the future. There is a wide range of research on the relationship between perceived value and satisfaction, and most conclude that the perceived value has a positive impact on customers’ satisfaction [44,45]. Most studies have also confirmed that perceived value has a positive impact on tourist satisfaction [46,47]. In the study of trust and satisfaction, trust is generally used as an intermediary variable to study the relationship between customers’ satisfaction and revisit intention. Some research results show that trust can directly and positively affect satisfaction [48,49]. In conclusion, the perceived value and satisfaction, trust and satisfaction are closely related. When the tourists’ perceived value increases, their satisfaction will also increase simultaneously, and tourist satisfaction is the result of increasing tourists’ trust. Therefore, this study puts forward the following hypotheses:

**Hypothesis** **4** **(H4).**
*Festival value has a positive impact on satisfaction with the festival.*


**Hypothesis** **5** **(H5).**
*Trust in the festival has a positive impact on satisfaction with the festival.*


### 2.6. Festival Loyalty

Gyte and Phelps [50] who were the first to propose the concept of revisit intention, found that British tourists have a revisit intention to Spain. Revisit intention refers to the tourists’ willingness to go to the destination they have been to again and to purchase its tourist products and services [51]. Academic research on satisfaction and revisit intention involves all industries that are rich in content. Most conclude that the two concepts can directly or indirectly have a positive impact [52,53,54]. To sum up, tourist satisfaction will affect the tourists’ revisit intention. If tourists are satisfied with the events and services of the destination, they are more likely to revisit and recommend the destination positively. Therefore, this study proposes the following hypothesis:

**Hypothesis** **6** **(H6).**
*Satisfaction with a festival has a positive impact on festival loyalty.*


## 3. Research Design and Method

### 3.1. A Case of Theme-Based Festivals

Within the current trend of international relations and economic development, Macau, as an international city where Chinese and Western cultures intersect, promotes and highlights its tourism destination image by hosting various theme-based festivals and events (e.g., music) and attracts tourists from all over the world. According to Macau Tourism Bureau, there are 23 events in Macau in 2019, including the annual Macau Food Festival. In 2010, Macau was awarded the honorary title of “the world culinary capital” by UNESCO. Since 2000, the annual Macau Food Festival has been successfully held 19 times. The 19th Macau Food Festival was given the theme “Celebrate the 20th anniversary of Macau reunification with world gourmet”, and it invited more than 150 famous catering companies to demonstrate the unique multi-cuisine culture of Macau and present a feast of “the world culinary capital” for tourists.

### 3.2. Research Architecture

The main purpose of this study is to understand the impact of service quality of the Macau Food Festival on tourists’ perceived value, trust, satisfaction, and revisit intention. According to previous studies, SERVQUAL service quality theory model should not be suitable for theme-based festivals. This study explores the attributes of theme-based festival quality, and whether service quality attributes may affect festival value, trust in the festival, satisfaction with the festival, and festival loyalty. After reviewing relevant literature, we utilize the activity service quality scale developed by Song et al. [8] to test the effect of festival quality of the Macau Food Festival on festival value, trust in the festival, satisfaction with the festival, and festival loyalty.

### 3.3. Research Hypotheses

Hospitality, venue, product, convenience, and program are taken as the first-order dimensions to measure festival quality. It is assumed that the second-order construct of festival quality may have an impact on tourists’ festival value, trust in the festival, satisfaction with festival, and festival loyalty.

The research hypothesis model is shown in Figure 1.

### 3.4. Questionnaire Design

Four items related to hospitality were adapted from Song et al. [7] and Song, Bae, and Lee [8]. Four items related to venue were adapted from Song et al. [7] and Song et al. [8]. Four items related to product were adapted from Song et al. [8]. Four items related to convenience were adapted from Song et al. [7] and Song et al. [8]. Finally, four items related to program were adapted from Yoo, Jin, and Choong [27] and Shu, Cole, and Charles [17]. Four items related to perceived value were adapted from Wong, Ji, and Liu [5] and Yang, Gu, and Cen [55]. Four items related to trust were adapted from Song et al. [7] and Song et al. [8]. Five items related to satisfaction were adapted from Song et al. [8], Aidin [9], and Faruk [10]. Four items related to revisit intention were adapted from Cevat, Bekir, and Alan [12], Appalayya and Justin [11], and Faruk [10]. In this study, multiple measurement items are employed to reduce the measurement errors such as common methods bias.

The current study uses 7-point Likert scales to score measurement items, ranging from completely disagree (1) to completely agree (7). As the main respondents are visitors from mainland China, the developed English questionnaire was translated into Chinese. Before sending out the questionnaire, the respondents were asked orally whether they had participated the Macau Food Festival. If the answer was yes, the investigation was continued. If the answer was no, the investigation was terminated. The questionnaire was divided into two parts: (a) the questions corresponding to each variable in the model hypothesis, using 7-point Likert scales and (b) the demographic characteristics of the respondents, including gender, age, and education level, the average household income, region, the professional background, and the number of times they had participated in the Macau Food Festival.

### 3.5. Sample Collection

From November 10th, 2019, to November 12th, 2019, the authors conducted a preliminary survey on the service quality, trust, satisfaction, and revisit intention of tourists participating in the 19th Macau Food Festival by simple random sampling. A total of 100 questionnaires were distributed, including 97 valid ones. According to the statistical software analysis, the Cronbach’s alpha of the pre-survey questionnaire is 0.966, greater than 0.8, which indicates very good reliability. In addition, during the pre-survey, when interviewing with the respondents, many respondents said that the item “parking convenience” could not be measured. Since most mainland tourists could not drive to Macau by themselves, and indeed, there was no parking lot on the site of Macau Food Festival, this item was revised to “the convenience of free shuttle bus”. The formal questionnaire was distributed from November 16th, 2019, to November 24th, 2019. The locations were selected at the site of the 19th Macau Food Festival and the nearest bus station. A total of 453 questionnaires were distributed by simple random sampling, and 433 valid questionnaires were recovered, with an effective rate of 95.58%.

## 4. Study Findings

### 4.1. Descriptive Analysis

According to the descriptive analysis of demographic data, women account for 56.8% of the respondents, aged 18–30 years (64.0%); the education level is mainly college / University, accounting for 64.4%; the occupation is mainly students, accounting for 38.8%; the annual household income range of the respondents is balanced; 60.0% of the respondents are from mainland China, 34.9% from Macau, China, 3.5% from Hong Kong, China, and Taiwan and overseas tourists accounted for 1.2% and 0.5% of the total; 188 (43.4%) respondents had participated in the Macau Food Festival once or twice, 96 (22.2%) had participated in 3 or 4 times, 17.6% had participated in the festival more than 5 times, and 16.9% had not participated in the festival. The result of demographic profile of respondents is shown in Table 1.

### 4.2. Reliability Analysis and Validity Test

Reliability is used to test the internal consistency and stability of the questionnaire results. Cronbach’s alpha and composite reliability are the most commonly used methods to measure the reliability. The lowest acceptable value of Cronbach’s α coefficient is 0.7, and if the value is higher, the reliability of the questionnaire will be better [56]. By using Smart-PLS statistical software, the α values of all variables in this study were found to range from 0.851 to 0.900, larger than the recommended value of 0.7, and the composite reliability (CR) coefficient’s value is between 0.903–0.933; that is to say, both Cronbach’s α value and CR value meet the standard value requirements. Validity analysis is used to test and measure the authenticity and accuracy of questionnaires. The commonly used methods include convergent validity (CV) and discriminant validity (DV). Convergent validity refers to the similarity of the results of the same variable measured by different questions, while discriminant validity refers to the difference between different variables [57]. The factor loading of all variables in this study is in the range of 0.768–0.914, which is greater than the recommended value of 0.7. At the same time, the average difference extraction rate (AVE) of each variable was between 0.692–0.778, which was higher than the recommended value of 0.5. In conclusion, the scale has good convergent validity. In Table 2, the square roots of the average difference extraction rate (AVE) values on the diagonal are higher than the correlation coefficients corresponding to other variables, indicating that the measurement scale of this study has good discriminant validity. In conclusion, the scale has good convergent validity (CV) and discriminant validity (DV) (See Table 3).

### 4.3. Structural Model and Hypotheses Test

There are nine variables in this study, including five attributes used as structural indicators to create the second-order factor of festival quality. The first-order variables of festival quality were analyzed to test the correlation between the five first-order variables and festival quality. Smartpls3.0 was used to analyze the five first-order variables and test the support degree of festival quality. Then, a bootstrapping method was employed to calculate the path coefficient and *t*-value of the first-order variables, which were used to measure the importance of the first-order factors corresponding to festival quality. All five variables were significantly associated with festival quality. Among these five paths, program was the most significant variable (path coefficient = 0.273, *t* value = 26.747), followed by product (path coefficient = 0.266, t value = 25.152), hospitality (path coefficient = 0.256, *t* value = 27.123), convenience (path coefficient is 0.238, *t* value is 22.121), and venue (path coefficient = 0.222, *t* value = 19.598). This study used Smart-PLS to calculate the path of the model. Prior to the first step, in the bootstrap setting of the final analysis attribute, the original data were re-extracted 5000 times. On this basis, the significance of the path coefficient was tested. According to the validation criteria proposed by Wong [58], when the *t*-value of the path is greater than 1.96 and the *p*-value is less than 0.05, the path coefficient of the model is significant. According to the results in Table 4, festival quality has a positive impact on festival value, supporting H1; festival quality has a positive impact on trust, supporting H2; festival value has a positive impact on trust in the festival, supporting H3; festival value has a positive impact on satisfaction, supporting H4; trust in the festival has a positive impact on satisfaction, supporting H5; andsatisfaction with the festival has a positive impact on revisit intention, supporting H6. All in all, this research model supports all hypotheses (See Table 5 and Figure 2).

## 5. Conclusions

Nowadays, festivals and events are attracting tourists rapidly worldwide, and the service quality is a vital factor to successfully holding festivals and events. According to the path analysis model, it can be seen that hospitality, venue, product, convenience, and program have a significant impact on the service quality of the Macau Food Festival. Among them, saliency is ranked from strong to weak as follows: program, product, hospitality, convenience, and venue. When discussing the relationship between festival quality, festival value, and trust in festival, it was found that festival quality positively affects festival value (path coefficient = 0.598, *t*-value = 13.768); festival quality positively affects trust in the festival (path coefficient = 0.361, *t*-value is 5.460); and festival value also positively affects trust in the festival (path coefficient = 0.386, *t*-value = 6.090) (See Table 5). This shows that good service quality for the event is conducive to improving the perceived value and trust level of tourists. Tourists have a high overall evaluation of festival quality, which helps tourists to form a psychological belief and enjoyment, so that the emotional value of the tourists is satisfied, which in turn affects the perceived value of the tourists in festival settings.

Festival value has a positive impact on trust in festival. If tourists believe that the Macau Food Festival is a festival with a wide variety of products, high traffic access, and rich experience events, and they can get pleasure and relaxation in the process of their participation, then they will get a better sense of the festival experience and then generate trust in it. Festival value has a positive effect on satisfaction with the festival (path coefficient = 0.415, *t*-value = 7.703), which shows that the higher the perceived value of tourists, the higher the evaluation of their perceived emotional value, social value, and functional value, and the more that visitors can feel the real satisfaction during the event.

The degree of trust in the festival has a positive effect on satisfaction with the festival (path coefficient = 0.320, *t*-value = 5.542). It shows that the higher the festival value, the higher their psychological expectations of the image and status of the festival, and the higher satisfaction with the festival. Paying attention to the level of festival trust and how to improve the level of tourists’ festival trust and thereby enhance the level of satisfaction with the festival are important issues that DMOs and festival hosts should pay attention to.

Satisfaction with the festival has a positive effect on festival loyalty (path coefficient = 0.636, *t*-value = 17.291). This shows that tourists are satisfied with the activity itself, and they are willing to recommend the festival to others. Therefore, for food festival events, it is important to focus on the improvement of the service quality for theme-based festivals, to find the factors that affect the satisfaction of tourists, and to improve the satisfaction of tourists, which is particularly important for improving festival loyalty.

### 5.1. Theoretical and Practical Implications

The current study found that hospitality, venue, product, convenience, and program are regarded as a bundle of festival quality in sufficient validity and reliability. The findings of this study support the view of Song et al. [7], proposing the significant role of festival quality in enhancing the magnitude of festival loyalty. Specifically, the results showed that the second-order factor of festival quality (consisting of program, product, hospitality, convenience, and venue) plays a critical role in facilitating the level of festival loyalty through festival value, trust in festival, and satisfaction with the festival. This implies that when the tourists perceive the good service quality for a theme-based festival, they perceive a positive sense of value and trust in the Macau Food Festival. Additionally, the higher their satisfaction is, the more satisfied the tourists are with festival loyalty. Then, they may participate in the theme-based festival (i.e., the Macau Food Festival) next year or later. To sum up, this study puts forward suggestions from five aspects—festival quality, festival value, trust in the festival, satisfaction with the festival, and festival loyalty—hoping to help the long-term development of other food festival events such as the Macau Food Festival.

Based on the findings of the empirical analyses, this study also offers some practical implications as follows.

First, business mode and service quality should be improved. In order to create joyful festival experiences for tourists, it is necessary not only to rely on the variety and authenticity of cuisine, but also the excellent service performance of the staff. The results of this study showed that hospitality has a positive impact on the service quality of the Macau Food Festival. Therefore, the service attitude and performance of the staff is an indispensable element to the improvement of the overall service quality.

Second, ingredients’ characteristics should be used properly and innovative products should be created. At the food festival, the most attractive thing to tourists is food. Tasting food is the most obvious purpose of tourists visiting the Macau Food Festival. From the study results, it is known that products have the greatest impact on the service quality for an event, so the variety, taste, and price of the food and beverages in the Macau Food Festival can affect tourists’ evaluation of the overall service quality. Therefore, under the premise of ensuring food safety, the Macau Food Festival should cook ingredients through special processing to make them full of color and taste. At the same time, the cuisine should be diversified to meet the different needs of tourists.

Third, traffic problems should be solved and convenience increased. The customers of the Macau Food Festival are mostly local residents and tourists from neighboring provinces and cities. For convenience, the Macau Food Festival provides visitors with four free shuttle bus routes. However, according to interviews with the tourists during the sampling period, tourists on weekends are less satisfied with the shuttle bus than tourists on weekdays, because the shuttle bus scheduling is difficult on Saturday and Sunday. The free shuttle bus scheduling problem during weekends can be solved by adjusting the bus from the other three routes to reduce the transportation pressure. Only when traffic problems are effectively improved can tourists participate in the activities with pleasure. Otherwise, the chaotic traffic conditions will not only affect the mood of tourists but also affect the overall service quality of tourists and even negative impact on the festival event.

Fourth, scientific and reasonable planning is necessary to enrich the program. The Macau Food Festival not only has many food stalls but also special show performances every night from 19:30 to 21:00, as well as “Slam Dunk”, “Happy Bowling”, and other events, which provide tourists with leisure and entertainment. Before planning and designing festivals and special events, the leisure and entertainment needs should be considered, and reserve appropriate leisure spaces for tourists between the crowded food stalls. At the same time, the rich experience of events can also make tourists feel happy and interested. In addition, performances inject more vitality into the Macau Food Festival, but the language of the show performance is Cantonese. Most mainland tourists do not understand, so it is recommended that the show performance can include some Mandarin programs every night. It is better to arrange some interactive programs that tourists can participate in to make them feel relaxed and improve their perceived interest.

Fifth, perceived value should receive focus on and tourist satisfaction should be increased. Perceived value is the tourists’ overall evaluation comparing psychological expectations before traveling and the actual experience in travel. This research measures the total tourists’ perceived value of the 19th Macau Food Festival through four items: “worthy of time, worthy of energy, worthy of cost, and worthy overall.” It can be known that perceived value can positively affect tourists’ satisfaction. No matter the purpose of the visit to the Macau Food Festival, visitors can have a higher perceived value after participation, and their satisfaction will also increase, which may eventually lead to tourists’ intention to revisit. Therefore, organization planners can investigate the perceived value and hire a professional research team to investigate and analyze the factors that affect the tourists’ perceived value and need to actively understand the difference in the perceived value of different demographic characteristics groups, so as to accurately improve the service quality, tourists’ perceived value, and satisfaction. In this way, it can not only be in an advantageous position in the market competition but also obtain satisfaction and revisit.

Sixth, the festival image should be built and tourists’ trust enhanced. The festival image refers to tourists’ subjective cognition and evaluation. Whether the festival image is good or bad will affect the tourists’ perceived value, satisfaction, and revisit intention. The “halo effect” is actually a deviation of cognition. When a person has a positive cognition of something, other aspects of the thing are also considered positive. For the Macau Food Festival, planners need to create a good festival image to attract more potential tourists, because tourists can have a positive understanding of the festival image during the participation process, and tourists’ trust will also be enhanced and their expectations will be met.

Seventh, tourists’ satisfaction and loyalty should be improved. Tourist satisfaction is the standard of evaluation of whether tourists are satisfied with the festival hosting destination. This research measures the tourists’ satisfaction with the 19th Macau Food Festival through five items: “decision satisfaction, product satisfaction, price satisfaction, environmental satisfaction, and overall satisfaction”. It can be concluded that satisfaction with the festival positively affects tourists’ festival loyalty. The higher the degree of satisfaction with the festival, the more likely they are to revisit. Moreover, revisiting tourists can not only generate income for the destination again but also increase the popularity of the destination through word-of-mouth and expanding the potential market. Therefore, the Macau Food Festival should pay attention to the factors that affect tourists’ satisfaction and firmly grasp the realization of the relationship between satisfaction with the festival and revisit intentions, to improve the tourists’ satisfaction and promote tourists’ festival loyalty.

### 5.2. Limitations and Future Research

Although the study has many useful theoretical and practical implications, there are also some limitations. First, due to limited human resources, only 453 questionnaires were distributed in this study, and 433 valid questionnaires were recovered. Therefore, whether the conclusion of this study represents other food festivals and events needs further demonstration. Second, a simple random sampling survey method was used, and the questionnaires were mainly distributed at the entrance of the food festival, the waiting area for the free shuttle bus, and the waiting place for a meal. The irritability of queuing for meals or waiting may have a negative impact on the service quality for the food festival. Based on the above research limitations, in future research, the field investigation time and the scope of respondents should be increased and expanded, so that it can be accurately understood whether the population of different demographic characteristics differs in the choice of each item and to make descriptive statistical analysis more objective and clear. A staged survey should be taken and samples selected at the beginning, middle, and later stages of the festival to conduct comprehensive research. In addition, this research explores the research model of service quality in terms of event, perceived value, trust, satisfaction, and revisit intention. The research only selected the single case of Macau Food Festival to test the model and did not extend it to other cases. However, there are many other types of festivals such as traditional, cultural, and sports events. The image and experience differ in various types of festivals. For example, tourists’ trust in a sports event may be higher than the perceived value, which may also lead to different results. It is hoped that future research can use the proper research model to measure other types of festivals and special events.

## Figures and Tables

**Figure 1 ijerph-18-09214-f001:**
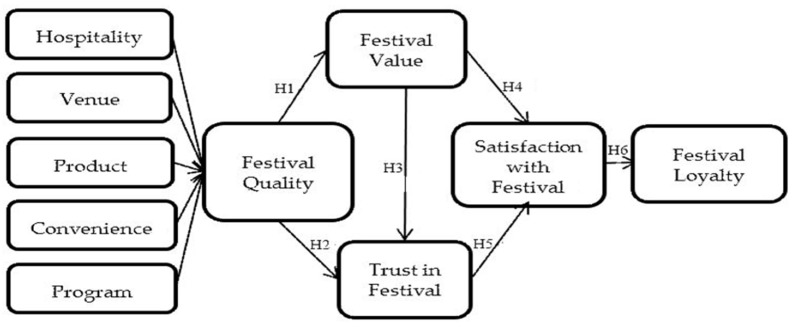
Research hypotheses.

**Figure 2 ijerph-18-09214-f002:**
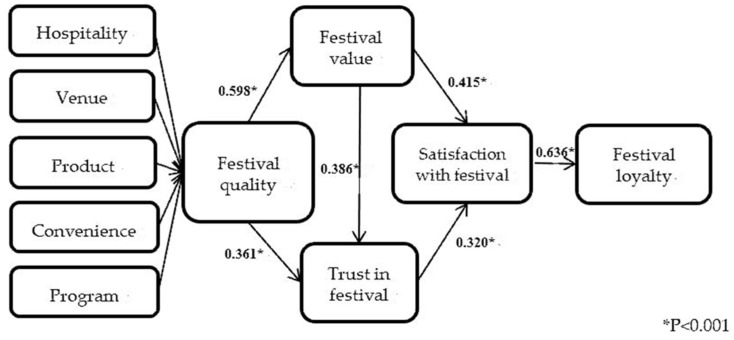
Structure model analysis of sample.

**Table 1 ijerph-18-09214-t001:** Demographic profile of Sample (*N* = 438).

Demographic	Frequency	(%)
Gender		
Female	246	56.8
Male	187	43.2
Age		
18–30	277	64.0
31–40	96	22.2
41–50	53	12.2
51 or above	7	1.6
Education		
High school or below high school	85	19.6
College degree	94	21.7
Bachelor degree	185	42.7
Master degree	61	14.1
Doctor degree	8	1.8
Average annual household income		
48,000 RMB/60,000 MOP and below	100	23.1
48,012 RMB/60,012 MOP-96,000 RMB/120,000 MOP	92	21.2
96,012 RMB/120,012 MOP-144,000 RMB/181,200 MOP	97	22.4
144,012 RMB/181,212 MOP-192,000 RMB/242,400 MOP	79	18.2
192,012 RMB/242,412 MOP and above	65	15.0
Ethnicity		
Chinese Mainland	260	60.0
Macau, China	151	34.9
Hong Kong, China	15	3.5
Taiwan, China	5	1.2
Other	2	0.5
Visit frequency of Macau Food Festival		
None	73	16.9
Once or twice	188	43.4
Three to five times	96	22.2
More than five times	76	17.6
Occupation		
Student	168	38.8
Housewife	23	5.3
Civil servants	3	0.7
Personnel engaged in service industry	54	12.5
Faculty personnel	18	4.2
Business personnel	25	5.8
Personnel engaged in manufacturing industry	9	2.1
Self-employed	63	14.5
Other	70	16.2

**Table 2 ijerph-18-09214-t002:** Confirmatory factor analysis for measurement model.

Items	Factor Loading	AVE	Composite Reliability
Hospitality			
The number of staff was properly equipped.	0.878	0.750	0.923
The staff had the ability to solve problems.	0.883		
The staff was polite.	0.856		
The staff had good verbal and nonverbal communication skills.	0.848		
Venue			
Venues’ facilities	0.862	0.704	0.905
Space and size of venues	0.851		
Cleanliness of the facilities in venues	0.819		
The atmosphere of venues	0.823		
Product			
Quality of related to products	0.861	0.737	0.918
Price of the products	0.853		
Taste of the products	0.876		
Variety of products	0.843		
Convenience			
Convenient public transportation from a point of departure to the festival venues.	0.867	0.699	0.903
Accessibility from to the festival area to other venues.	0.858		
Easy access to the Festival area	0.845		
Convenience in taking the free shuttle bus.	0.771		
Program			
Program was funny.	0.846	0.763	0.928
Program was varied	0.83		
Experiential program was wonderful	0.866		
Program was well managed	0.846		
Festival Value			
Compared to time I spend, attending the festival is worthy.	0.878	0.761	0.927
Compared to the efforts I made, attending the festival is worthy.	0.893		
Compared to the expenses I paid, attending the festival is worthy.	0.845		
Overall, I feel that attending the festival is worth the money.	0.873		
Trust in Festival			
Macau gained my confidence as a region for food and tourism destination by hosting the festival.	0.883	0.772	0.931
Hosting the festival improved Macau’s brand image for food and tourism destination.	0.882		
Hosting the festival strengthened my belief in Macau as a food and tourism destination	0.868		
Hosting the festival in Macau met my expectations of it as a food resort and as a tourism destination.	0.881		
Satisfaction with Festival			
I am satisfied with my decision to visit the festival.	0.825	0.711	0.925
I am satisfied with food taste at the festival.	0.834		
I am satisfied with product price at the festival.	0.845		
I am satisfied with dining environment at the festival.	0.831		
I am satisfied with the overall service quality of the festival overall.	0.879		
Festival Loyalty			
I will come to the festival again next year.	0.911	0.777	0.933
If had to decide again I would choose the festival again.	0.911		
I would come back to the festival in the future.	0.896		
The festival would be my first choice over other theme-based festivals.	0.803		

**Table 3 ijerph-18-09214-t003:** Analysis of discriminant validity.

Items	1	2	3	4	5	6	7	8	9
1.Hospitality	0.865								
2. Venue.	0.592	0.832							
3.Product	0.542	0.498	0.858						
4.Convenience	0.524	0.446	0.563	0.836					
5. Program	0.544	0.495	0.611	0.594	0.873				
6. Festival Value	0.443	0.388	0.548	0.469	0.518	0.872			
7. Trust in Festival	0.407	0.423	0.557	0.480	0.480	0.602	0.877		
8. Satisfaction with Festival	0.328	0.294	0.465	0.399	0.428	0.608	0.570	0.842	
9. Festival Loyalty	0.365	0.340	0.427	0.401	0.394	0.474	0.538	0.636	0.882

**Table 4 ijerph-18-09214-t004:** First-order path analysis.

Path	Path Coefficient	*t-*Value
Hospitality–Festival quality	0.256	27.123
Venue–Festival quality	0.222	19.598
Product–Festival quality	0.266	25.152
Convenience–Festival quality	0.238	22.121
Program–Festival quality	0.273	26.747

**Table 5 ijerph-18-09214-t005:** Smart-PLS path test results.

Path	Path Coefficient	*t-*Value	Hypothesis
H1: Festival quality→Festival value	0.598	13.768	Supported
H2: Festival quality→Trust in festival	0.361	5.46	Supported
H3: Festival value→Trust in festival	0.386	6.09	Supported
H4: Festival value→Satisfaction with festival	0.415	7.703	Supported
H5: Trust in festival→Satisfaction with festival	0.32	5.542	Supported
H6: Satisfaction with festival→Festival loyalty	0.636	17.291	Supported

## Data Availability

Can be provided from the corresponding authors upon reasonable request.

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
