# Peer review of "Examining the Role of Service Quality, Perceived Values, and Trust in Macau Food Festival"

_ijerph, 2021, doi:10.3390/ijerph18179214_

Round 1

Reviewer 1 Report

I believe the overall quality of the paper is good, however I think it does not bring any new highlight or new find to this academic field. Somehow, the conclusions were a bit predictable. Despite that, I believe the paper is well written, well-structured, and well supported.

Author Response

Dear Reviewer,

We highly appreciate your review of this paper. 

Based on other reviewers' advice, we have revised and updated this paper.

Reviewer 2 Report

Detailed comments

The article presents an interesting study on the role of service quality based on  Macau Food Festival case study. The paper is generally well structured and meets the standards of scientific papers. The statistical analysis and results are clearly presented. Please, find enclosed the detailed comments that are meant to improve the quality of the presented study. 

Author Response

Dear Reviewer,

We highly appreciate your review of this paper.

Based on your advice and other reviewers' tips, we have revised and updated this paper.

See below

The article presents an interesting study on the role of service quality based on  Macau Food Festival case study. The paper is generally well structured and meets the standards of scientific papers. The statistical analysis and results are clearly presented. Please, find enclosed the detailed comments that are meant to improve the quality of the presented study. 

 Detailed comments

The article presents an interesting study on the role of service quality based on Macau Food Festival case study. The paper is generally well structured and meets the standards of scientific papers. The statistical analysis and results are clearly presented.

The following comments are meant to improve the quality of the presented study:

  1. Abstract – A brief description of the research method employed is missing. The Authors indicate Questionnaires, which are research tools, not methods.

Answer: Out of 453 questionnaires distributed, 433 questionnaires were valid for data analysis using PLS-SEM (partial least squares-structural equation modelling).

There is a typo (lines 13-14): “[…] and its indirect impacts on festival loyalty through festival value, trust in festival, satisfaction with festival, and festival satisfaction.”

Answer: and its indirect impacts on festival loyalty through festival value, trust in festival, and visitors’ satisfaction. We have edited this part.

  1. Keywords: Authors used too many (three to ten pertinent keywords were required) and they do not seem to be proper: “festival with satisfaction”? Or rather: “customers’ satisfaction.” Another ambiguous term is “festival loyalty” (lines: 13, 22,51,54, 203, 213, 233, 236, 241, 319, 321, 348, 379, 384, 389, 395, 398, 468, 474). Perhaps, the Authors could reconsider using “customers’ loyalty towards festival” or “revisit intention” instead.

Answer: As you mentioned, we have changed like this.

Keywords: Macau food festival; festival quality; festival value; trust in festival; visitors’ satisfaction; revisit intention

  1. The Introduction requires some changes. It lacks clearly in terms of the relationships among variables, and hypotheses to be tested.

Answer: We have explained this part: Festival linked with food has tended to ignore its festival quality and value. As a case of this research, the annual Macau Food Festival was selected to explore visitors' perceptions on festival quality and its indirect impacts on festival loyalty through festival value, trust in festival, and visitors’ satisfaction.

In lines 51-55 Authors stated: “Therefore, the current study aims to explore the relationships between festival quality, festival value including the five sub-dimensions of hospitality, venue, product convenience, and program, trust in festival, satisfaction with festival, and festival loyalty using partial least squares structural equation modelling (PLS-SEM) analyses”.

It may be assumed that hospitality, venue, product, convenience, and program are the festival sub-dimensions, while in the body of the paper Authors present them as festival quality dimensions (sub-dimensions) – see lines 76 – 153.

The relationship is clearly stated in lines 228-229.

Answer: Therefore, the current study aims to explore the relationships among festival quality (five sub-dimensions of hospitality, venue, product convenience, and program), festival value, trust in festival, satisfaction with festival, and festival loyalty.

  1. As the festival quality is the main variable, Authors should present the concept of service quality in more detailed way, indicating different approaches and quality dimensions referred to by other researchers, and clearly indicate why they chose hospitality, venue, product, convenience, and program.

Answer: From a psychological perspective, Gronroos [20] coined the concept of perceived service quality as an index for detecting the degree of personal perceptions toward service experiences. Based on this theoretical foundation, the SERVQUAL scale [21] and the SERVPERF scale [22] were developed and utilized to evaluate the level of ex-pected service experiences in hospitality and tourism research. However, it would not be possible for researchers to measure the magnitude of service quality for festivals using the two scales because individuals’ expected experiences at a special interest fes-tival would be more detailed than other service experiences from hotels, restaurants, and destinations (e.g., [23]). In this regard, some studies have verified that the scale of service quality for event including hospitality, venue, product, convenience, and pro-gram should be employed to predict visitors’ cognitions such as perceived value, trust, and satisfaction in the formation of festival loyalty [24-26]. In this regard, theme-based festival quality can be measured by the attributes of hospitality, venue, product, con-venience, and program in festival settings. The concepts of each festival quality uti-lized in this study are summarized as follows:

  1. In 3.5 sample collection (line 270) there is a statement: “From November 10th, 2019, to November 12th, 2019, the author conducted a preliminary survey […]”- was there only 1 Author? Or Authors?

Answer: From November 10th, 2019 to November 12th, 2019, the authors conducted a pre-liminary survey on the service quality, trust, satisfaction and revisit intention of tour-ists participating in the 19th Macau Food Festival by simple random sampling. – We have edited this one.

  1. The Discussion part is missing, Authors used “Conclusions” instead. It is not the same and should not be mixed.

Discussion: Authors should discuss the results and how they can be interpreted in perspective of previous studies and of the working hypotheses. The findings and their implications should be discussed in the broadest context possible, and limitations of the work highlighted. Future research directions may also be mentioned. This section may be combined with Results.

Conclusions: This section is mandatory and should provide readers with a brief summary of the main conclusions (see Instruction for Authors).

Answer: Since we have written theoretical and practical implication and limitation and future research under conclusion and discussion, we wish these comprises both of them.  5. Conclusion and discussion

  1. More up-to-dated publications should be referred to (e.g. 2020, 2021).

Answer: We have checked and updated some of articles recent version ones.

  1. Proofreading would be required to avoid typos.

Answer: Thank you for your tips and we have checked once again with professional editors.

Reviewer 3 Report

Interesting article that analyzes the study of the impact of an event – the Macau Food Festival. However, the number of inquiries appears to be insufficient. In addition, the geographic location of this study needs to be better valued. Macau is a complex tourist territory based on the game and some universal heritage, of Portuguese origin. In this sense, it would be interesting to know a little more about this Macau Food Festival and its real connections to the Macanese territory.

Author Response

(The authors gave the same response as above.)

Reviewer 4 Report

Dear Authors,

The study aims at investigating the "The role of service quality for event perceived value, trust, satisfaction, and revisit intention".

The paper is not satisfactory written, needs a careful editing, fonts, and style.

Further, the study aim and background are not well presented, repetitions occurring in the paper should be avoided.

However, it is recommended:

- Reformulate the abstract by telling prospective readers what you did and what the important findings of your research were. I suggest not to use acronyms in the abstract.

- Introduction can be improved in order to show better aim. Is very short.

- Please carefully consider and revise the logic of some parts.

- Carefully check the full text.

- Literature coverage in terms of papers is not balanced, references are classic. In terms of literature review, the paper does, unfortunately, take no unique or critical point of view. The paper should better provide a critical analysis of the available and appropriate literature.

- I suggest considering a general / integrative theoretical approach in order to present your research model, and then write your paper from the angle of the chosen one.

- Please do more to highlight how the work advances or increments the field from the present state of knowledge and provide a clear justification for your work.

- The methodological approach used is not clear. Seems like a statistical exercise not an econometric approach.

- Results must be rewritten.

- Conclusion section needs improvement. Please provide more quantitative key contributions of the study with proper discussions, highlight the limitations of this study and the future work.

- English proofreading is needed. Some description is not professional for a scientific article.

Accordingly, it is opinion of this reviewer to accept with major revisions the proposed manuscript for a publication on this journal.

Author Response

Dear Reviewer,

We highly appreciate your review of this paper.

Based on other reviewers' advice, we have revised and updated this paper.

The study aims at investigating the "The role of service quality for event perceived value, trust, satisfaction, and revisit intention".

The paper is not satisfactory written, needs a careful editing, fonts, and style.

Answer: Examining the role of service quality, perceived values, and trust in Macau Food Festival

Further, the study aim and background are not well presented, repetitions occurring in the paper should be avoided.

Answer: We have tried to explain the study aim and background with these explanations. As an index for predicting future demand in an annual theme-based festival, exploring how festival visitors form festival loyalty in a cognitive process is highlighted by many scholars due to its critical role in detecting if festival visitors are willing to be repeated visitors. The necessity of understating festival loyalty calls for a deeper investiga-tion of what factors will be directly and indirectly affect festival loyalty in a cognitive process model. Given this recognition, many studies have proposed the prominent at-tributes of service quality and identified their impacts on loyalty.

However, it is recommended:

- Reformulate the abstract by telling prospective readers what you did and what the important findings of your research were. I suggest not to use acronyms in the abstract.

Answer: Based on your advice, we have deleted acronyms in the abstract.

- Introduction can be improved in order to show better aim. Is very short.

Answer: We have tried to explain the study aim and background with these explanations. As an index for predicting future demand in an annual theme-based festival, exploring how festival visitors form festival loyalty in a cognitive process is highlighted by many scholars due to its critical role in detecting if festival visitors are willing to be repeated visitors. The necessity of understating festival loyalty calls for a deeper investiga-tion of what factors will be directly and indirectly affect festival loyalty in a cognitive process model. Given this recognition, many studies have proposed the prominent at-tributes of service quality and identified their impacts on loyalty.

- Please carefully consider and revise the logic of some parts.

Answer: Based on your advice, we have checked and changed some formats for better logic of this paper.

- Carefully check the full text.

- Literature coverage in terms of papers is not balanced, references are classic. In terms of literature review, the paper does, unfortunately, take no unique or critical point of view. The paper should better provide a critical analysis of the available and appropriate literature.

Answer: We appreciate your advice about literature review and we added more information of Macau Food Festival.

- I suggest considering a general / integrative theoretical approach in order to present your research model, and then write your paper from the angle of the chosen one.

Answer: Based on your advice, we have updated research model. 

- Please do more to highlight how the work advances or increments the field from the present state of knowledge and provide a clear justification for your work.

Answer: Based on your advice, we have added more information for clear justification.

- The methodological approach used is not clear. Seems like a statistical exercise not an econometric approach.

- Results must be rewritten.

Answer: Based on your advice, we have checked once again and rewritten methodological approach.

- Conclusion section needs improvement. Please provide more quantitative key contributions of the study with proper discussions, highlight the limitations of this study and the future work.

- English proofreading is needed. Some description is not professional for a scientific article.

Answer: Thank you for your tips and we have checked once again with professional editors.

Accordingly, it is opinion of this reviewer to accept with major revisions the proposed manuscript for a publication on this journal.

Answer: We appreciate your considerable editing opinion and we have tried our best for better quality of paper.

Round 2

Reviewer 4 Report

Dear Authors,

I very much appreciate the efforts of the authors to meet my comments and suggestions and to implement the suggestions, observations and recommendations I made. The paper has improved but there are however some major aspects that need to be still fixed and to which no answer was given. 
The authors claim to have improved the methodology and conclusions, but I don't see any difference from the previous version.
Similar consideration for improving the literature...It's still 62 references as in the previous version. Accordingly, it is opinion of this reviewer to accept with major revisions the proposed manuscript for a publication on this journal.

Author Response

Dear Reviewer, 

First of all, we really appreciate your concern about this paper and we have checked this paper and tried to revise based on your advice. Regarding references, we have already enough references with this paper so hope you understand this part. Once again, we really appreciate your comments. 
